# Do Vegetarian Diets Provide Adequate Nutrient Intake during Complementary Feeding? A Systematic Review

**DOI:** 10.3390/nu14173591

**Published:** 2022-08-31

**Authors:** Giovanni Simeone, Marcello Bergamini, Maria Carmen Verga, Barbara Cuomo, Giuseppe D’Antonio, Iride Dello Iacono, Dora Di Mauro, Francesco Di Mauro, Giuseppe Di Mauro, Lucia Leonardi, Vito Leonardo Miniello, Filomena Palma, Immacolata Scotese, Giovanna Tezza, Andrea Vania, Margherita Caroli

**Affiliations:** 1AUSL Brindisi 1, ASL Brindisi, 72023 Mesagne, BR, Italy; 2Department of Primary Cares, AUSL Ferrara, 44121 Ferrara, FE, Italy; 3Health District 63, ASL Salerno, 84019 Vietri Sul Mare, SA, Italy; 4Department of Pediatrics, Belcolle Hospital, 01010 Viterbo, VT, Italy; 5Independent Researcher, 84100 Salerno, SA, Italy; 6Independent Researcher, 82100 Benevento, BN, Italy; 7Department of Primary Cares, AUSL Modena, 41012 Carpi, MO, Italy; 8Health District 19, ASL Caserta 2, 81038 Trentola Ducenta, CE, Italy; 9Health District 17, ASL Caserta, 81031 Aversa, CE, Italy; 10Maternal Infantile and Urological Sciences Department, Sapienza University, 00161 Rome, RM, Italy; 11Nutrition Unit, Department of Pediatrics, “Giovanni XXIII” Children Hospital, “Aldo Moro” University of Bari, 70126 Bari, BA, Italy; 12Health District 65, ASL Salerno, 84091 Battipaglia, SA, Italy; 13Helath District 64, ASL Salerno, 84022 Campagna, SA, Italy; 14Department of Pediatrics, San Bortolo Hospital, 36100 Vicenza, VI, Italy; 15Independent Researcher, 00162 Rome, RM, Italy; 16Independent Researcher, 72021 Francavilla Fontana, BR, Italy

**Keywords:** complementary feeding, weaning, vegetarian, growth, overweight, neurodevelopment, malnutrition, vitamin B12, infections, not communicable diseases

## Abstract

During the complementary feeding period, any nutritional deficiencies may negatively impact infant growth and neurodevelopment. A healthy diet containing all essential nutrients is strongly recommended by the WHO during infancy. Because vegetarian diets are becoming increasingly popular in many industrialized countries, some parents ask the pediatrician for a vegetarian diet, partially or entirely free of animal-source foods, for their children from an early age. This systematic review aims to evaluate the evidence on how vegetarian complementary feeding impacts infant growth, neurodevelopment, risk of wasted and/or stunted growth, overweight and obesity. The SR was registered with PROSPERO 2021 (CRD 42021273592). A comprehensive search strategy was adopted to search and find all relevant studies. For ethical reasons, there are no interventional studies assessing the impact of non-supplemented vegetarian/vegan diets on the physical and neurocognitive development of children, but there are numerous studies that have analyzed the effects of dietary deficiencies on individual nutrients. Based on current evidence, vegetarian and vegan diets during the complementary feeding period have not been shown to be safe, and the current best evidence suggests that the risk of critical micronutrient deficiencies or insufficiencies and growth retardation is high: they may result in significantly different outcomes in neuropsychological development and growth when compared with a healthy omnivorous diet such as the Mediterranean Diet. There are also no data documenting the protective effect of vegetarian or vegan diets against communicable diseases in children aged 6 months to 2–3 years.

## 1. Introduction

An adequate and balanced diet is an essential to achieving and maintaining healthy status. According to the World Health Organization (WHO), approximately one third of cardiovascular and neoplastic diseases could be prevented by a healthy diet [1], which could also serve as a protective factor against mental disorders [2].

A healthy diet needs to provide on its own, without any supplements, all the macro- and micronutrients to fully satisfy, at any stage of infancy and childhood, all nutritional needs and to promote the best possible psychophysical development. A healthy diet must also be a protective factor against the widest possible range of diseases while at the same time preventing the development of specific nutrient-deficiency or excess conditions. Therefore, a healthy diet must include a varied, balanced intake of foods, with all food groups being represented in the right proportions in terms of quantity and frequency of consumption [3].

It is important to underline that no diet considered healthy and recommended by the WHO excludes food groups. Specifically, no diet excluding meat and other foods of animal origin is recommended.

In Western countries, vegetarianism has become particularly popular in recent years. Nevertheless, to date, few reliable data exist on the prevalence of vegetarian eating habits in Western countries, due to unclear and different definitions of vegetarianism (e.g., not eating meat does not automatically exclude eating fish) and surveys conducted with incomparable protocols [4]. 

In the US, data from the National Health and Nutrition Examination Survey (NHANES) Data Report (2007–2010) show a 2.3% prevalence of vegetarian adults among the general population [5,6]. 

Despite uncertainties about the true prevalence of vegetarianism, the number of people opting for a vegetarian diet appears to be gradually increasing also in Europe, albeit with somewhat fluctuating data. 

The number of people in the UK claiming to be vegetarian has increased dramatically over the last half-century; statistics from the Second World War suggest that 0.2% of the population were vegetarian in the 1940s, and it is estimated that, in 2000, between 3% and 7% of the population had switched to vegetarianism [7]. In the 2003–2006 period in Germany, approximately 2.1% of the boys and 6.1% of the girls interviewed (aged between 14 and 17) reported following a vegetarian diet [8]. According to data collected by the Eurispes in 2022 [9], 5.4% of Italians claim to be vegetarian, in addition to 9.7% of people who although no longer vegetarians claim to have been vegetarian in the past; a remaining 1.3% claim to be vegan. From 2014 to date, albeit with some fluctuations, the number of vegetarians seems to have slightly decreased. 

Since individuals in the pediatric age group tend to follow their family’s dietary patterns, it may be assumed that the prevalence of infants, children and adolescents following vegetarian diets is also increasing [10].

The studies aimed at exploring the reasons for vegetarianism have found a number of motivations, both religious and non-religious [4], such as health and ethical reasons [11,12,13], a moral obligation not to harm animals [14], environmental concerns [9,15] and specific disgust for meat [16,17].

Vegetarian diets can be classified as pescetarian, lacto-ovo vegetarian (LOV), lacto-vegetarian, ovo-vegetarian and vegan. These forms can have specific subtypes, such as: raw food diet, fruitarian diet, macrobiotic diet, etc. [4,7,8,9,10,11,12,13,16,17,18,19,20,21,22,23] (Table 1).

Nutritional quality indices such as the Healthy Eating Index (HEI) [24], the Mediterranean Diet Score (MDS) [25] and the Alternate Healthy Eating Index (AHEI) [26,27] are used to measure dietary health by assessing different dietary patterns within an omnivorous nutritional model. None of these questionnaires excludes a priori any food groups.

Indeed, diets with important restrictions carry the risk of not meeting nutritional requirements in terms of both energy and nutrient intake, notably in more fragile physiological situations such as pregnancy, lactation and early childhood.

Several nutrition societies support the use of vegetarian diets at all life stages, requiring, however, nutrient supplementation when needed. According to the Academy of Nutrition and Dietetics (USA) [18], the Australian National Health and Medical Research Council [19] and the Portuguese National Program for the Promotion of a Healthy Diet [20], well-planned vegetarian diets are appropriate for individuals during all life stages, including pregnancy, lactation, infancy, childhood and adolescence. The Canadian Paediatric Society [21] also states that well-planned vegetarian and vegan diets, with the use of food supplements, can meet the nutritional needs of children and adolescents. According to the British Nutrition Foundation [7] a well-planned and balanced vegetarian or vegan diet may be nutritionally adequate and provide the nutrients necessary to grow up healthy. 

Contrary to these positions, the German Nutrition Society (DGE) [22] recommends a diet that includes all food groups, including animal-based products, especially during pregnancy, lactation and childhood. DGE also strongly states that in case a vegetarian diet is followed, supplementation, food selection, and medical monitoring are essential for the early detection of any nutrient deficiencies. A few years ago, the Italian Society of Preventive and Social Pediatrics (SIPPS), together with the Italian Federation of Pediatricians (FIMP) and the Italian Society of Perinatal Medicine (SIMP), published a position paper on vegetarian diets from pregnancy to adolescence, delving into the issue of the appropriateness of these diets in terms of children’s growth and neurocognitive development and also exploring their effects on communicable and not-communicable diseases and eating disorders [23]. This position paper concludes that a vegan diet cannot be recommended for children because it leads to deficiencies in vitamin B12, DHA, iron, vitamin D and calcium. Whenever such a diet is proposed, it must absolutely be supplemented with all the above-mentioned nutrients. Moreover, children following any other type of vegetarian diet may present with nutritional deficiencies and must be carefully monitored in their growth and general development. 

It is now more common that vegetarian parents consult the pediatrician for advice on how to provide their infants with a CF partially or totally free of animal-source foods. Notably, the first 1000 days are particularly vulnerable from a metabolic and neurodevelopmental point of view [10]. Therefore, a systematic review (SR) of the literature on the possible outcomes of vegetarian diets during CF was considered of paramount importance.

### 1.1. Why This Systematic Review Is Important

A pitfall of scientific literature is that simplified definitions of vegetarian diet are often reported. As a result, the real diet composition adopted in some studies might be uncertain. An example is misreported cases of semi-vegetarian or pescetarian diets (incorrectly referred to as vegetarian) that mask nutritional deficiencies regularly detected in the most extreme vegetarian diets including vegan diet and LOV diets. Moreover, when evaluating a study, breastfeeding and food supplementation must be considered as they might strongly impact pediatric outcomes including growth and neurodevelopment.

Furthermore, it often happens that vegetarian dietary patterns regarded as valid alternatives are checked against the so-called Western Diet, whose compositional features and health effects appear to be extremely different from those recognized as valid by the WHO such as the Mediterranean Diet (MD). 

Focusing on the CF period, the available literature data mostly concern exclusively breastfed infants over the first 6 months of life and vegetarian mothers who were hospitalized for anemia, rickets and/or neurocognitive regression related to vitamin B12, vitamin D and/or iron deficiency.

Finally, to date, only a few SRs have been published focusing on vegetarian complementary feeding outcomes. Therefore, this SR is focused on the evaluation of these types of diet in this specific age group, and it takes into account the methodological limitations of the studies analyzed.

### 1.2. Objectives

The aim of this SR is to answer to some clinical questions concerning the influence of vegetarian diets during CF on the development of the following outcomes: growth, neurodevelopment, risk of wasted and/or stunted growth, overweight, obesity.

Additional outcome(s) are risk of deficiency of vitamins and micronutrients, infections, developing Type 2 Diabetes Mellitus (T2DM) and hypertension later in life.

### 1.3. Key Questions

Does complementary feeding completely or partially free of animal-source foods in healthy infants result in significantly different auxological development and/or growth compared with that of infants on a balanced omnivorous diet?Does complementary feeding completely or partially free of animal-source foods in healthy infants/toddlers result in a psychomotor development that is significantly different from that of infants on a balanced omnivorous diet?Do healthy infants on a complementary feeding that is completely or partially free of any animal-based foods
◦have a different risk of developing NCDs (obesity/overweight, hypertension, diabetes mellitus)?◦have a different risk of developing vitamin or other micronutrient deficiencies?◦have a different risk of developing infections and related outcomes compared with infants on a balanced omnivorous diet?


The questions, structured according to PICO, are reported in Appendix A.

## 2. Materials and Methods

Details of the protocol for this SR were registered with PROSPERO and can be accessed at PROSPERO 2021 CRD 42021273592 [28].

The formulation of clinical questions, the search, the analysis of the scientific evidence with specific tools [29,30,31,32], and the GRADE method [33,34,35] have been previously described [36].

### 2.1. Design of the Studies Included

The studies included were as follows:Randomized controlled trials (RCTs) and controlled trials (CTs) in which the effect of the caregivers’ feeding practices could be accurately assessed as an experimental intervention.Observational studies (cohort studies, longitudinal studies, case-control studies, and cross-sectional studies) in which this effect could be evaluated as an exposure factor while taking into account possible confounding factors.

To improve bibliographic search resources and to obtain further elements for the evaluation of the studies included in this SR, evidence-based guidelines or consensus statements, government publications and already published SRs considered to be of good quality were also selected and evaluated.

### 2.2. Population

Healthy, term born, normal birth weight, breastfed and/or formula-fed infants ages 6–24 months or older for long-term outcomes, residing in Western industrialized countries.

The context is the primary care in Western industrialized countries

### 2.3. Intervention(s), Exposure(s)

Complementary feeding (CF) with small quantities or without any animal product (vegetarian or vegan diet).

### 2.4. Comparator(s)/Control

Complementary feeding (CF) that includes food of animal origin.

### 2.5. Inclusion Criteria

Intervention and observational studies conducted in industrialized countries.Studies in which the intervention or exposure was present during the period of complementary feeding (6–24 months).Studies comparing vegetarian diets (LOV, vegan, macrobiotic and others, completely free of animal protein) and healthy diets (e.g., MD).Studies with follow-up greater than 12 months.

### 2.6. Exclusion Criteria

Studies conducted on populations with characteristics different from those established in the PICOs (e.g., children living in LICs (low-income countries), preterm infants, low birth weight infants, children who developed peri-neonatal diseases and children with chronic diseases).Non-comparative studies.Comparisons between vegetarian and Western-type diets or other non-healthy diets are excluded as wholly inappropriate.Studies on so-called dietary patterns in the absence of a precise definition/documentation of what in the individual study was considered a “vegetarian pattern or diet”.Follow-up of fewer than 12 months.Case reports and case series, despite low quality, have been included in this SR, being a large part of the literature on this topic.

Studies with methodological biases likely to affect the confidence in the results obtained were also excluded [37,38].

### 2.7. Outcomes

#### 2.7.1. Main Outcomes

General growth parameters assessed in prospective differential terms (different increase in weight (W) or length (L) over time) or assessed at a specific time point (with differing frequencies of weights and lengths in the populations being compared: W, L, W/L z-score ratio, body mass index (BMI), BMI z-score (BMIz).Risk of wasted and/or stunted growth.Risk of impaired neurodevelopment.Risk of NCDs (overweight/obesity).

#### 2.7.2. Additional Outcomes

Risk of deficiency of vitamins and micronutrients.Risk of infections.Risk of NCDs (T2DM, and hypertension) later in life.

### 2.8. Keywords and Search Strategy

See Appendix A.

### 2.9. Measures of Effect

The standard methods of the Cochrane Review Group were used to synthesize the data; the effects have been reported as risk ratio (RR), odds ratio (OR), risk difference (RD) with 95% confidence intervals (CIs) for categorical data and as mean difference (MD) or standardized main difference (SDM) (95% CIs) for continuous data [36].

### 2.10. Study Selection

Study Selection: risk of bias (quality) assessment; missing data; list of the studies excluded with relevant reasons; data extraction (selection and coding). See Appendix A.

The selection process and the assessment of the overall methodological quality and of the quality of the individual studies, including missing data handling and data extraction, were conducted by at least two mutually blind authors, as already reported [36].

The SRs and the studies were evaluated using specific tools [30,31,32], as previously described [36].

The following data were extracted from the studies: author, year of publication, study design, objective of the study, country, sample, healthy or pathological condition, age, type of intervention, period of follow-up, results, main conclusions of the study and financing.

### 2.11. Strategy for Data Synthesis. Additional Analysis of the Results

It was not possible to perform meta-analysis because it was not possible to merge the data of at least 2 studies. In this SR, additional analyses of the results were made, when possible, to calculate effect measures (RR, OR) not reported in the studies, important for the purpose of evaluating the effect of vegetarian diets on the outcomes considered (Appendix A).

### 2.12. Software

The software RevMan 5.4.1 was used to report the quality of RCTs and present the results graphically [39].

The GRADEpro GDT software, developed by the GRADE Working Group, was used to grade the overall quality of evidence and the related tables (Appendix A) [40].

## 3. Results

### 3.1. Does Complementary Feeding Completely or Partially Free of Animal-Source Foods in Healthy Infants Result in Significantly Different Auxological Development and/or Growth Compared with That of Infants on a Balanced Omnivorous Diet?

This SR included the following studies and documents: a multidisciplinary and evidence-based document that makes graded recommendations relevant to the question [23] (Figure 1), an SR of low methodological quality [41] (Figure 2) from which the only relevant study of moderate quality was selected [42]. In addition, a cross-sectional study of moderate quality was included [43] (Figure 3).

The Position Paper SIPPS–FIMP–SIMA–SIMP [23] states the following:

“Given the very low level of evidence, it is not possible to state with certainty that vegetarian diets in childhood and adolescence ensure adequate growth and nutritional status.

Given the very low level of evidence, it is not possible to establish at what age a vegetarian diet can be started without side effects on growth.

Quite the contrary, evidence does exist on the need to supplement diets that exclude certain food groups. The more restricted a diet is, the greater nutritional deficiencies are.

Specific nutritional counseling is recommended for appropriate supplementation, with particular reference to the amino acid profile of proteins and intakes of iron, zinc, vitamin B12 and DHA (strong positive recommendation).

Periodic nutritional status assessments including the prescribed supplementation in both children and adolescents are recommended (strong positive recommendation)”.

The quality of the evidence is very low, as only four observational studies were included, and none of them answered the question. Indirect data from studies conducted on children during breastfeeding or after the start of CF, or in early childhood, or at later ages were therefore included. Despite the very low quality of the evidence, the recommendations made in this paper were ‘strong positive’ because of the great importance of the documented harmful effects of non-supplemented restrictive diets.

With regard to meat intake, the SR by English et al. [41] reports that “twelve articles, seven RCTs (references in the SR nr. 2, 3, 5, 11, 14, 15, 18) and five articles from four observational studies (references in the SR n. 32, 33, 39, 44, 49) examined the relationship between meat intake and growth, size, body composition, and/or risk of overweight or obesity outcomes. Studies varied in terms of whether they assessed meat compared to cereal, meat and cereal compared to controls, or the amount of meat consumed”. The studies compared groups of infants who consumed meat or cereal in various amounts and of different qualities, or different amounts of meat but were not really relevant to our question based on the set of inclusion criteria of this SR (studies conducted in developed countries, age of exposure, diet to which controls were exposed, period of observation and length of follow-up). A specific exposure to a vegetarian diet is reported in only one of these studies [42]. The above-mentioned study is an observational cohort study of moderate methodological quality in which the control diet is not properly described in detail in the group defined as omnivorous (OM), and the confounding factors for the adjusted analysis of the results are not given. A group of 49 children, from 4 to 18 months of age, on a macrobiotic diet free of animal protein was compared with a group of 57 OM children that did not differ significantly in month of birth, sex, parity, education of the parents and region of residence. Within the macrobiotic group, growth differences, expressed as change in SD scores (SDS) per year, were found between the ages of 8 and 14 months for weight velocity: 3.3 vs. 4.4 kg/y—*p* < 0.001; length velocity, 13.2 vs. 16.7 cm/y—*p* < 0.001; head circumference velocity, 5.2 vs. 6.1 cm/y—*p* < 0.05.

These differences disappeared at 18 months, but in the follow-up study carried out 2 years later [44], children from families who had increased their consumption of animal products since the start of the study grew in height faster than the other children (*p* < 0.05).

A significantly higher risk was also observed of skin and muscle wasting in 30% of macrobiotic infants compared with 2% of the controls (*p* < 0.001), RR (95%CI) = 17.45 (2.39, 127.38), *p* = 0.005 (Appendix A).

The VeChi Diet Study, a cross-sectional study of moderate quality, was also selected for this SR [43].

The energy and macronutrient intake and the anthropometric data (average weight-for-height, height-for-age and weight-for-age z-score) of 430 children aged 1–3 years (127 vegetarian (VG), 139 vegan (VN) and 164 OM) were analyzed.

There were no significant differences in energy intake and average anthropometric measures between the study groups. 

However, in more detail, some significant differences did exist, and the intake of some nutrients was not balanced across the three types of diet.

OM children had the highest adjusted median intakes of protein (OM: 2.7, VG: 2.3, VN: 2.4 g/kg body weight, *p* < 0.0001), but it was emphasized that on average all groups had 2.3–2.5-fold higher protein intake than the German reference value (1 g protein/kg BW and day).

OM children had the highest adjusted median intakes of fat (OM: 36.0, VG: 33.5, VN: 31.2%E, *p* < 0.0001) and added sugars (OM: 5.3, VG: 4.5, VN: 3.8%E, *p* = 0.002), whereas VN children had the highest adjusted intakes of carbohydrates (OM: 50.1, VG: 54.1, VN: 56.2%E, *p* < 0.0001) and fiber (OM: 12.2, VG: 16.5, VN: 21.8 g/1000 kcal, *p* < 0.0001).

The VN group, thus, had the highest carbohydrate intake but was particularly unbalanced in the amount of fiber (10 g/1000 kcal being the attainable value) [45].

In the study, all diet groups exceeded this value, which was, however, significantly higher in the VG group (unadjusted median [interquartile range, IQR] = 16.1 [13.8–20.0]) and even higher in the VN group (19.6 [16.3–24.1]) even with very high intakes (30–45 g/day). 

As to the anthropometric data, a higher prevalence of children with inadequate growth was observed in the vegetarian and vegan diet groups: 21/266 VN and VG children were classified as stunted, i.e., with insufficient energy and long-term nutrient intake, or classified as wasted, with severe malnutrition, compared with 1/164 OM children (OR (95% CI) = 13.97 (1.86, 104.88); *p* = 0.01) (Appendix A).

The summary of findings for the main comparisons is shown in Table 2.

### 3.2. Does Complementary Feeding Completely or Partially Free of Animal-Source Foods in Healthy Infants/Toddlers Result in a Psychomotor Development That Is Significantly Different from That of Infants on a Balanced Omnivorous Diet?

To improve the comprehensiveness of the research in the specific area of neuropsychological disorders, it was decided to extend the search to the PsycINFO database of the American Psychological Association (APA).

From the search for evidence-based LGs, only one document was selected that made recommendations relevant to the question [36]. The position paper reports that there are no data available on the safety of CF without animal-based foods; the scientific evidence only consists of case reports or case series. It recommends close nutritional monitoring of the infant, even after the start of CF, with supplementation as needed to avoid severe clinical outcomes such as growth deficits, anemia and especially neurological deficits (strong positive recommendation).

It is important to emphasize that all the documents, even those not included in this SR, irrespective of their methodological quality, diet patterns adopted and purpose, recommend that the intake of vitamin B12, iron, zinc and DHA be adequate to meet infant needs, as deficiencies in these micronutrients in the early stages of the development of the CNS can cause severe and irreversible damage.

No relevant SRs were found. Available evidence, searched and retrieved without time limits, is limited to the previously mentioned cohort study by Dagnelie et al. [42] and to case reports or case series focused on deficiencies in individual nutrients, which although relevant do not include a control arm and thus rank at the lower end of the efficacy evidence scale (Section 3.3.2).

In addition to the effects on physical growth already reported, the study conducted by Dagnelie et al. also assesses, in the same groups, the effects of exposure to animal protein-free macrobiotic diets on certain parameters of psychomotor development. In that study, child development differences were expressed in SD scores, with negative values indicating slower development.

The macrobiotic group was significantly slower in gross motor and, to a lesser degree, in speech and language development. The difference was greatest in locomotion:Gross Motor DevelopmentSitting balance and head control SD = −0.48; *p* = 0.04.Walking SD = −0.60; *p* = 0.001.Overall score SD = −0.63; *p* < 0.001.

Fine Motor Development SD = −0.13; *p* = 0.49.Language Development SD = −0.42; *p* = 0.03.

The babies in the macrobiotic diet group reached independent walking on average 3 months later than the OM babies; the toddlers in the same group with low muscle mass and adipose tissue exhibited tendentially slower motor development (*p* = 0.05). Delayed and lower speech development was correlated, only in the macrobiotic group, with low-birth weight (*p* < 0.05).

A number of case reports provide data on the effects of vegetarian diets on psychomotor development (Appendix A) [46,47,48,49,50,51,52].

Appendix A summarizes 10 cases of children aged 8–18 months: All of them exhibit severe neurological outcomes and growth deficits resulting from low vitamin B12 and vitamin D levels, with anemia, stunting, brain abnormalities and demyelination. Cases with persistent outcomes are reported with no long-term follow-up data for any other disorders.

The summary of findings for the main comparisons is shown in Table 3.

### 3.3. Do Healthy Infants on a Complementary Feeding That Is Completely or Partially Free of Any Animal-Based Foods

Have a different risk of developing NCDs (Obesity/Overweight/Hypertension, Diabetes Mellitus)?Have a different risk of developing vitamin or other micronutrient deficiencies?Have a different risk of developing infections and related outcomes compared to infants on a balanced omnivorous diet?

#### 3.3.1. Risk of Developing NCDs

The systematic search for any GLs in the past five years that could be relevant to the clinical question did not lead to any results.

The search for SRs in the last 10 years did not yield any relevant results.

The results showed no significant difference: 36/266 children on a vegetarian/vegan diet were overweight or obese compared to 23/164 children on an omnivorous diet. (OR (95%CI) = 0.96 (0.55, 1.69); *p* = 0.89) (Appendix A).

The search for original studies did not identify any study on the influence of vegetarian diets during CF on the risk of developing hypertension or T2DM later in life.

#### 3.3.2. Risk of Developing Vitamin or Other Micronutrient Deficiencies

Only one paper was selected as it provides recommendations relevant to the question [23]. The position paper reports that scientific evidence only consists of case reports or case series, essentially of infants older than 6 months breastfed by vegetarian or vegan mothers, admitted to the emergency department with several neurological symptoms and receiving a well-documented diagnosis of micronutrient deficiencies.

From the search for evidence for this SR, two comparative cohort studies were also selected, both of moderate quality [42,53].

Also in the study conducted by Dagnelie et al. [42], children on a macrobiotic diet developed deficiencies of energy, protein, vitamin B12, vitamin D, calcium and riboflavin, leading to retarded growth, fat and muscle wasting and slower psychomotor development.

In spite of higher iron intake, iron deficiency was observed in 15% of macrobiotic infants, but not in controls (*p* = 0.003).

Plasma vitamin B12 concentrations were 149 pmol/L (geometrical mean) in macrobiotic infants compared with 404 pmol/L in the control group (*p* < 0.001).

Mean folate concentrations were higher in the macrobiotic group, being considered an effect of vitamin B12 deficiency.

Low riboflavin intake was reflected in biochemical evidence of riboflavin deficiency. The activity coefficient of erythrocyte glutathione reductase (EGR) was 1.14 ± 0.10 (± SD) in the macrobiotic group compared with 1.01 ± 0.06 in the control group (*p* < 0.001). One quarter, 26%, of the macrobiotic infants had an EGR activity coefficient > 1.20, compared with 2% of the control infants (*p* < 0.001).

Finally, plasma 25-hydroxyvitamin D (*p* < 0.001), calcium (*p* < 0.05) and phosphate (*p* < 0.01) concentrations in summer were significantly lower in macrobiotic infants than in control infants. The proportion of macrobiotic with major clinical rickets (five physical symptoms of rickets, of which two were specific-rickets-related) increased from 4% in summer to 45% in winter (*p* < 0.001).

Taylor et al. [53] evaluated whether the iron and micronutrient status of children aged 4–24 months improved as soon as meat intake increased. They compared 150 children (198–48 lost at follow-up) divided into 2 groups on the basis of their diet pattern: non-meat eaters (*n* = 20) or mixed (red and white)-meat eaters sub-grouped into tertiles depending on the meat content reported in diet diaries.

As for parameters of iron status, the number of results below the reference range was determined for each diet group, and a significant negative relationship between serum iron (<9.0 μmol/L) and meat intake at 12 months of age was seen (*p* < 0.023). There was an unexpected trend for hemoglobin concentrations to be inversely related to the meat intake, although not statistically significant, at the same age (*p* < 0.068). No effects on zinc or copper status were seen. However, it must be emphasized that the article does not specify whether the vegetarian group ate foods supplemented with iron.

Finally, seven articles were included for a total of ten case reports [46,47,48,49,50,51,52]. While the quality of evidence is low, the case reports are all consistent in demonstrating severe neurological outcomes and growth deficits in children on a vegan diet, due to low vitamin B12 and vitamin D levels, with anemia, stunting, brain abnormalities and demyelination. The children with vitamin B12 deficiency [46,47,48,49,50,51,52] all had psychomotor developmental deficits or neurological symptoms [46,47,48,49,50,51,52], some even anemia [48,49,50,51,52]. In one case, in addition to psychomotor regression, four bone fractures were found due to nutritional rickets [47]. The reason was prolonged breastfeeding from a vegetarian mother followed by a vegan diet for the infant after weaning. Treatment with vitamin B12, or vitamin D in case of rickets, relieved most signs and symptoms, except in cases where persistent clinical deficits were reported. No long-term follow-up data are available for any other disorders, e.g., learning disabilities (Appendix A).

#### 3.3.3. Risk of Developing Infections

No relevant SRs over the past 10 years or original studies in the past 40 years were identified in the search on the effects of vegetarian diets during CF on the risk of developing infections.

Only one position paper [23] was included that states that all available scientific evidence concerns only infants fed soy formulas. No evidence exists on older children on vegetarian diets, and therefore, the paper concludes that with regard to the risk of communicable diseases in children fed other types of plant-based formulas, such as rice formulas, or in older children, in the absence of evidence on a safety profile, vegetarian/vegan diets can neither be recommended nor discouraged.

## 4. Discussion

The CF period is characterized by rapid growth where infants are vulnerable to nutrient deficiencies and excesses. For this reason, the WHO recommends that infants receive a healthy diet containing all essential nutrients for growth and proper psychomotor development.

The growing popularity of vegetarianism among young populations in industrialized countries has resulted in an increasing number of parents asking pediatricians for a CF partially or totally free of animal foods for their infants [7]. As a result, significant interest exists in the potential impacts vegetarian or vegan diets exert on a range of health outcomes, such as growth and neurocognitive development, as well as of the effects these diets have as an exposure factor for communicable and noncommunicable diseases (in terms of both risk and prevention).

The objective of this SR was to verify whether the recommendations of the 2017 Italian position paper [23], which assessed the appropriateness of vegetarian/vegan diets for all stages of development from pregnancy through adolescence, were still valid.

### 4.1. Scientific Evidence on the Safety and Efficacy of Vegetarian Diets in Children and Adolescents

#### 4.1.1. Weight–Length Gain

The Italian Position Paper on Vegetarian Diets in Pregnancy and Developmental Age [23] answered the question of whether CF free of animal products represents a risk factor for reduced physical growth and development compared with that of children on a balanced omnivorous diet. The paper concluded that due to the very low level of scientific evidence, which consisted only of case reports or case series, many of which referred to infants older than 6 months and still exclusively breastfed, it is not possible to state with certainty that infants on a vegetarian diet exhibit a different growth pattern compared with infants on a diet including animal products. Nor it is possible to determine at what age a vegetarian diet can be introduced with no adverse side effects on growth. On the contrary, there is strong evidence that diets that exclude certain food groups need to be supplemented because nutritional deficiencies are all the greater the more restrictive a diet is. Therefore, because of the well-documented detrimental effects of unsupplemented restrictive diets, the position paper recommended that infants on CF with vegetarian/vegan diets undergo specific nutritional counseling for appropriate supplementation. Where the infant is fed a diet deficient in essential nutrients, thereby requiring supplementation, the natural function of nutrition is lost as food medicalization occurs.

The 2017 Italian Position Paper was followed by sparse additional scientific evidence on the topic. The SR by English et al., of low methodological quality, which included 49 studies, explored the relationships between types and amounts of complementary foods/beverages and growth, size and body composition.

Of these studies, the only relevant study to our question about the adequacy of vegetarian/vegan diets in CF is that by Dagnelie et al. [42].

This already mentioned cohort study has major limitations: a small sample population, and inappropriately detailed diets, generically defined as “omnivorous” or “adequate” and “inadequate”. In addition, the SR by English et al. reports that no conclusions can be drawn on the relationships between distinct dietary patterns during the CF period and growth and body size (Grade Not Assignable).

In a different report, Weder et al. [43] analyzed energy and macronutrient intake and anthropometric data in the groups of children described above. The study, also of moderate quality, has some limitations, including the cross-sectional design and the indirect collection of anthropometric data. Furthermore, the article does not report whether the diets were supplemented or not.

#### 4.1.2. Psychomotor Development

Nutritional deficiencies impair early brain development and function and often result in reduced myelination, dendritic arborization and synaptic connectivity that occur at a very early age [51]. The functional consequences of these alterations vary depending on the specific nutritional deficiency and on the time when it occurs with respect to the neurological processes in development. If an early environmental event reduces neural plasticity via epigenetic mechanisms, there is no guarantee that the removal of the specific negative stimulus will result in complete return to the previous condition. This is confirmed by the fact that the early diagnosis and prompt treatment of early iron deficiency successfully treat anemia but do not prevent all long-term neurological disabilities linked to this condition [54].

The possible risks from exposure to vegetarian diets at pediatric age span from fetal life to adolescence but are particularly high during the first two years of life. In diets without meat, fish and meat products, the most critically deficient nutrient is vitamin B12, but deficiencies in calcium, iron, iodine, zinc, and selenium, essential amino acids, ω-3 LC-PUFAs (EPA and DHA) and vitamins (riboflavin, vitamin D) may also occur [54,55,56,57,58,59,60,61].

Very few studies are available on the long-term effects of early deficiencies from vegetarian diets. Additional research is required in this field to investigate potential irreversible effects, such as motor or cognitive delays, as well as possible links to psychopathological disorders. At present, this latter relationship remains unproven, and the multifactorial nature of mental disorders does not allow any cause–effect links to be established between vegetarian diets and the alterations in psychoaffective development later in life.

The 2017 Italian position paper [36] reports that no safety data on vegetarian or vegan CF exist and that the quality of scientific evidence is very low; therefore, the close monitoring of the child’s growth and neuro-psycho-motor development, as well as appropriate supplementation are recommended. In the absence of SRs and rigorous studies, the evidence is limited to case reports or case series on single nutrient deficiencies with no control arm. The cohort study by Dagnelie et al. [42], while showing different and delayed psychomotor development in the group that was on the macrobiotic diet, does not allow for any conclusive results. It should be mentioned, however, that even the ten case reports [46,47,48,49,50,51,52], while presenting low-quality evidence, are all consistent in demonstrating severe neurological outcomes due to low vitamin B12 and vitamin D levels, with anemia, stunting, brain anomalies and demyelination (Appendix A) that are frequently not fully reversible.

#### 4.1.3. Risk of NCDs (Obesity/Overweight, Hypertension, Diabetes Mellitus)

In the position paper coordinated by Caroli et al. [23], a systematic review of the relevant literature highlighted the limitations of these studies in demonstrating the effect of various types of vegetarian diets on these outcomes. These studies are heavily biased, being mostly conducted on adult populations and presenting results that are therefore difficult to translate to a pediatric population.

Vegetarian diets were found to be effective on some surrogate outcomes, such as reductions in serum cholesterol and LDL but not in serum triglycerides and with some mixed results for HDL cholesterol. The same diets were also observed to be effective in reducing oxidative stress and body fat tissue. However, in the overall evaluation of these results, it should be taken into account that vegetarians have an overall healthier lifestyle with fewer risk factors (no alcohol consumption, cigarette, sedentary lifestyle, etc.) (very low quality of evidence).

Regarding the effect on hypertension, the clinical importance of these results was quite modest. While vegetarian diets are associated with lower BMI and lower risk of obesity, the decrease in blood pressure cannot be justified solely based on this factor. Vegetarian diets are often rich in potassium and polyunsaturated fatty acids and are associated with an overall healthier lifestyle, without cigarette smoking and alcohol consumption. There were no adjustments for these confounding factors in the analyses of the observational studies (very low quality of evidence).

The effectiveness of vegetarian diets in the prevention and treatment of T2DM in adult patients, even compared with omnivorous diets recommended for this condition, is confirmed. However, diabetics also have a tendency to refuse vegetarian diets because they are considered too restrictive (moderate quality of evidence). No data were available on the pediatric population.

The update of the position paper published on 2017 did not identify any study on the influence of vegetarian diets during CF focused on the risk of developing hypertension or T2DM later in life.

Finally, although vegetarian diets are often recommended as a prevention tool for overweight and obesity, in the only study relevant to the question, the VeChi Diet Study [43], the meta-analysis of the results found no protective effect of vegetarian diets in the early years of life, at least at a two-year follow-up (Appendix A).

#### 4.1.4. Risk of Vitamin or Other Micronutrient Deficiencies

In the position papers coordinated by Caroli et al. [23], the main micronutrients that were likely to be deficient in children on vegetarian diets were identified with their associated risk of developing various diseases.

Vegetarian children, and particularly vegans, due to lower absorption of non-heme iron, require higher iron intake (1.8 times that of omnivores), although the absorption of non-heme iron may be facilitated by meal composition, reducing phytate and polyphenol content and increasing that of vitamin C.

In vegan children, on the other hand, daily calcium intake may be insufficient for infants, particularly when CF is introduced, because the calcium content of breast milk, unaffected by a vegan diet, is no longer enough to meet infants’ needs. Moreover, infants and children up to 3 years of age who are on a vegan diet are at risk of deficiency in vitamin A and especially vitamin B12, which only occurs in foods of animal origin.

These data are also confirmed in this SR: The results from studies by Dagnelie et al. [42] and by Taylor et al. [53] do in fact report that in children on diets without animal foods, deficiencies in vitamin B12, vitamin D, calcium, iron and riboflavin lead to slower growth, fat and muscle wasting and slower psychomotor development.

Vitamin D deficiency, although present in almost all dietary patterns, is of greater importance in vegetarian diets, with the potential to lead to major clinical rickets [42], even with the risk of pathological fractures [47].

#### 4.1.5. Risk of Infection

There are no data on the effect of vegetarian diets during CF on the risk of developing infections. The search for SRs over the past 10 years and studies in the past 40 years did not identify any relevant studies.

In the position paper coordinated by Caroli et al. [23], scientific evidence concerned only infants fed soy formulas. Based on the results from four cohort studies and one RCT of moderate methodological quality, feeding soy formulas to infants was not a risk factor for communicable diseases. However, no evidence was reported on rice hydrolyzed protein formulas or on older children on vegetarian diets.

Given the composition of soy-based formulas, which are nutritionally balanced and fortified with vitamins and other micronutrients, these findings cannot be automatically translated to children who eat foods of varying composition and in varying amounts via CF.

### 4.2. Assessment of Vegetarian Diets: Main Issues

Ethical/religious/ecological/economic needs are at play when it comes to vegetarian diets, but the purpose of this SR was the evaluation of these diets exclusively from the perspective of a health determinant in developmental age, putting aside social aspects of the various diet choices. From a systematic review of the literature performed in the 2017 position paper by SIPPS–FIMP–SIMA–SIMP [23], major issues emerged about the approach and design of most papers investigating vegetarian diets.

#### 4.2.1. The Definition of a Healthy Diet

One problem that emerges in the debate on vegetarian diets is the definition of “healthy” diet. The often abused term “healthy diet” cannot be considered correct if such diet is “properly planned and supplemented”.

To define a diet as “healthy”, it is necessary that the diet itself succeed on its own in meeting the nutritional needs of the population under consideration. If the diet does not meet such constraints but requires additions or supplementation, it should be correctly defined as a “deficient diet in need of addition and/or supplementation”.

Therefore, in this SR, the supplementation given or not given to children on a vegetarian diet, if reported, was considered, among other factors, in the evaluation of the results.

#### 4.2.2. The Quality of Scientific Evidence

The second issue is the quality of the studies.

The WHO, the World Cancer Research Fund and the American Institute for Cancer Research recommend an omnivorous diet, both to meet nutritional needs and to prevent noncommunicable diseases. Moreover, they specify that for this purpose, it is not necessary to completely or in part exclude any food groups, including those of animal origin.

Thus, in order to propose a diet free of certain foods, it is first necessary to provide evidence of its safety (deficiency outcomes arising from absence of nutrients) by means of rigorous studies in order to demonstrate any advantages versus recommended omnivorous diets that are documented to be healthy, such as the MD.

Unfortunately, most studies come almost invariably with major biases, the most important being the following:Limited numbers of studies specific to some ages, particularly in the CF period.Lack of robust, evidence-based GLs where recommendations are not supported by reliable evidence.Lack of high-quality comparative studies, even if observational, and lack of RCTs in pediatric age.Results too often referred to intakes rather than robust outcomes such as growth and risk of developing conditions (e.g., anemia, overweight/obesity).Small samples or outdated cross-sectional studies;Unclear differentiation between different types of diets (non-pure VG or VN), as there is often uncertainty on the exposure factors (incorrect definition of “vegetarian diet”, supplementation, diet not strictly followed with occasional consumption of foods of animal origin);Studies, more often than not, relying on self-reported data;Comparative study with unbalanced omnivorous diets (Western diet) or not precisely defined (generically “omnivorous”), resulting in unclear results that are not transferable to patients with healthy eating habits.

#### 4.2.3. Supplementation: Compliance and Costs

A further problem in the evaluation of vegetarian diets as a healthy choice is the need for the supplementation of certain micronutrients such as vitamin B12, iron, zinc and DHA in varying amounts depending on the greater or lesser dietary restriction of foods of animal origin, age and specific requirements resulting from special conditions such as disease, sports activities.

It is beyond dispute that the best source of vitamins and minerals is a balanced and varied diet combined with outdoor physical exercise [62,63]. A healthy individual should need supplementation only in exceptional cases and times. All additional supplementations are therefore considered prescriptions that medicalize a natural function such as nutrition, without taking into account at least two critical issues: compliance and cost.

Compliance. Long-term compliance in the secondary prevention of recurrences and complications of chronic diseases, e.g., asthma, is difficult to achieve [64,65]. When there is no perception of risk because the subject is apparently healthy, long-term preventive medicine as necessary for vegetarian diets is hardly practiced. If a pediatrician has a vegetarian family among his or her patients, he or she must keep this in mind, as this exposes the child at risk of both short- and long-term negative outcomes.

Costs. Even in countries with universal health coverage and social welfare, these supplements are paid for by the citizens, and in a family in which there are several members adopting a vegetarian diet, especially if they are children, adolescents, pregnant or breastfeeding women, the purchase of these products for months and years can be expensive and not affordable for everyone.

Additional costs to be taken into account are the cost of nutritional counseling, medical checks and additional follow-up examinations, which are not needed by those who follow a healthy omnivorous diet. Staying healthy while being on a vegetarian diet is affordable mostly by middle- to high-income households.

## 5. Quality of Evidence

The quality of the evidence from the studies included in this SR was low: In fact, the methodological assessment of the observational studies with the GRADE method started at a low level.

In addition, from the evaluation of the studies with NOSs [32], there are further biases due to:The uncertainty of exposure (self-reported diet), and/orThe unclear definition of the comparison diet, generically described as “omnivorous”;The time of assessment of the outcome, and/orThe absence of the outcome of interest at the start of the study is not demonstrated.

Accordingly, the quality of the evidence was further downgraded to very low. Furthermore, for some outcomes (neurodevelopment, vitamin and micronutrient deficiency), the quality of the evidence is very low, as it consists only of case reports.

## 6. Agreements and Disagreements with Other Studies or Reviews

The update provided by this SR essentially confirmed the results of previous SRs.

The SR of English et al. [41] reports that “Grade was Not Assignable” and “No conclusion could be made about the relationship between distinct dietary patterns during the CF period and growth, size, body composition, and/or prevalence/incidence of malnutrition, overweight or obesity”.

However, a more careful analysis and a more articulated reflection on the results that takes into account the importance of the outcomes, the context and some fundamental ethical principles in medicine lead the authors of this SR to draw different conclusions.

## 7. Limitations of the SR and Potential Bias in the Review Process

A comprehensive search strategy was adopted to search and find all relevant studies.

This SR adopted very selective inclusion criteria.

For this reason, the number of studies included in this SR was smaller, but there was greater transferability of the results to our population of reference.

In Appendix A, the GLs, the SRs and the studies excluded with reasons are reported.

Different outcome measures were used in the studies included, thus making it not always possible to merge the results, some of which only appeared in narrative form.

The publication bias could not be assessed because of the small number of studies included in the analyses.

## 8. Implications for Research

Future studies are certainly needed in order to evaluate the impacts of different diets (vegetarians or omnivorous) during CF. These studies should contemplate the following:A design including the most important confounding factors: any supplements and breast/formula feeding. This will support the reliability of the results obtained, as well as the real impact of the type of diet on relevant outcomes.A clear definition of the exposure (in terms of type of diet), which should be limited to the sole period of CF (i.e., 6 months to 2 years of age) and be carefully monitored over time to ensure their real and continuous presence.Strict criteria to define which categories of infants and families can be enrolled as control groups; this will avoid similar expositions in subjects pertaining to different groups, as well as differences that might influence the results (e.g., different percentages of breastfed infants between the intervention and control groups).An appropriate follow-up period of time, possibly of at least three years, to collect data on predefined outcomes.The most limited drop-off possible, even in observational studies.A uniform instrumental documentation of specific outcomes, namely the anthropometric ones, that should be collected by qualified health care professionals.

## 9. Conclusions

For obvious ethical reasons, there are no interventional studies assessing the impact of non-supplemented vegetarian/vegan diets on the physical and neurocognitive development of children. On the contrary, there are numerous studies that have analyzed the effects of dietary deficiencies of individual nutrients.

From these studies, it can be deduced that vegetarian and vegan diets are inadequate for the correct neuro-psycho-motor development of children. In particular, deficiencies in vitamin B12, DHA and iron can cause damage to the nervous system, sometimes irreversible. This is well documented in the numerous clinical cases published in the literature. If possible, these supplements should begin during pregnancy planning, in the peri-conceptional period.

Based on current evidence, vegetarian and vegan diets during the CF period have no preventive effects on NCDs and CDs and may result in significantly different outcomes on neuropsychological development and growth when compared with a healthy omnivorous diet such as MD.

There are also no data documenting the protective effect of vegetarian or vegan diets against communicable diseases in children aged 6 months to 2–3 years.

In conclusion, the effects of vegetarian diets on communicable and not communicable diseases prevention are still largely undocumented.

Vegetarian diets have not been shown to be safe, and the current best evidence suggests that the risk of critical micronutrient deficiencies or insufficiencies and growth retardation is high. If a vegetarian or vegan diet is recommended by a pediatrician during the CF period, potentially serious side effects caused by vitamin and micronutrient deficiencies on growth and development must be considered very carefully.

As a consequence, vegetarian and vegan diets cannot be recommended during the CF period because of potentially serious side effects caused by vitamin and micronutrient deficiencies on growth and neurodevelopment.

## Figures and Tables

**Figure 1 nutrients-14-03591-f001:**
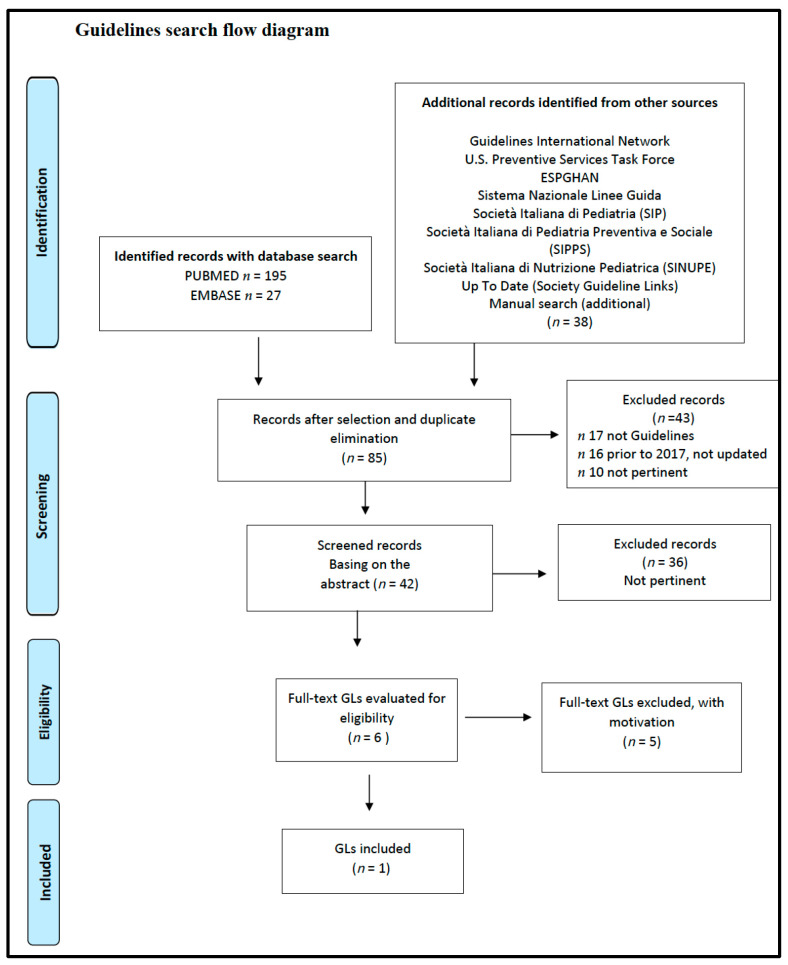
Vegetarian diets and different auxological development and/or growth. Flow diagram of the guidelines search. GL: Guide-Line.

**Figure 2 nutrients-14-03591-f002:**
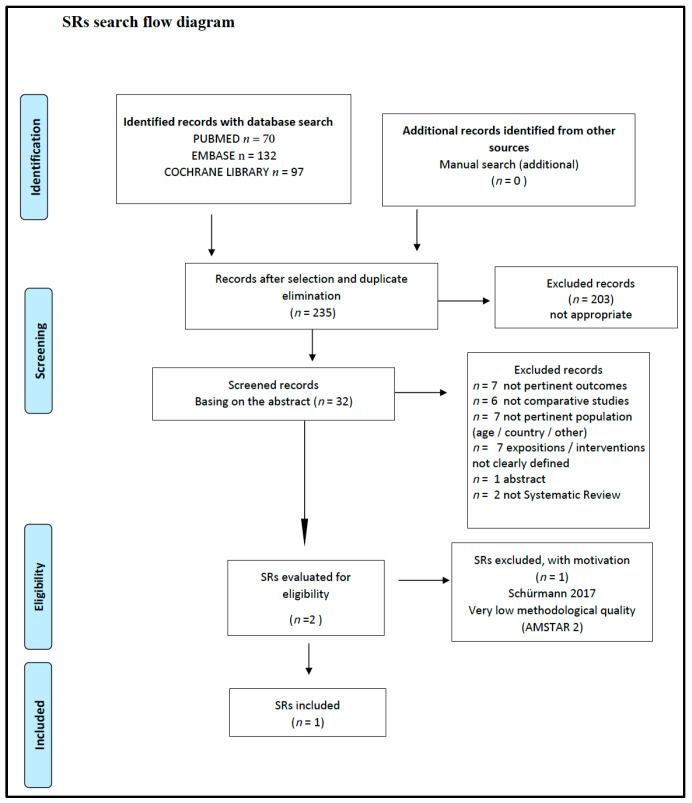
Vegetarian diets and different auxological development and/or growth. Flow diagram of the SRs search. SR: Systematic Review.

**Figure 3 nutrients-14-03591-f003:**
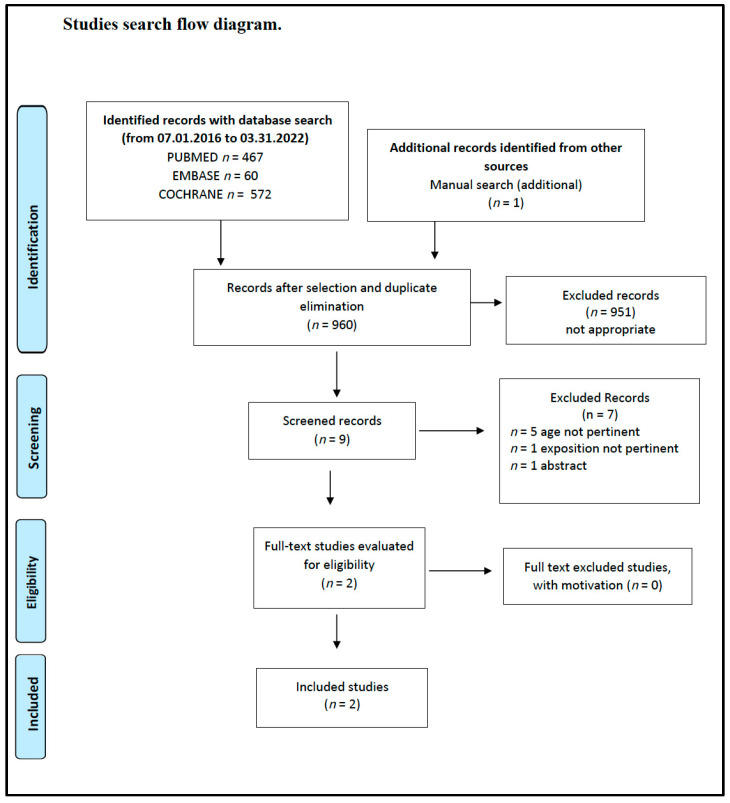
Vegetarian diets and different auxological development and/or growth. Flow diagram of the study search.

**Table 1 nutrients-14-03591-t001:** Vegetarian-type diets.

Diet Definition	Food Not Allowed	Food Allowed
**Pescatarian/pescotarian**	Meat (terrestrial animals, birds)	Fish, mollusks, crustaceans, seafood. Any plant-based food: cereals, legumes, vegetables, fruit, algae. Eggs, milk, dairy products, honey, royal jelly ^§^, propolis ^§^, mushrooms, yeasts, lactic ferments and brewer’s yeast
**Lacto-ovo vegetarian**	Meat, fish, mollusks	Any plant-based food: cereals, legumes, vegetables, fruit, algae. Eggs, milk, dairy products, honey, royal jelly ^§^, propolis ^§^. Mushrooms, yeasts, lactic ferments and brewer’s yeast
**Lacto-vegetarian**	Meat, fish, mollusks, crustaceans, milk and dairy products	Any plant-based food: cereals, legumes, vegetables, fruit, algae. Milk, dairy products, honey, royal jelly ^§^, propolis ^§^. Mushrooms, yeasts, lactic ferments and brewer’s yeast
**Ovo-vegetarian**	Meat, fish, mollusks, crustaceans, milk and dairy products	Any plant-based food: cereals, legumes, vegetables, fruit, algae. Eggs, honey, royal jelly ^§^, propolis ^§^. Mushrooms, yeasts, lactic ferments and brewer’s yeast
**Vegan**	All foods of animal origin, including: eggs, honey, milk and dairy products, propolis, royal jelly	Any plant-based food: cereals, legumes, vegetables, fruit, algae. Mushrooms, yeasts.
**Raw vegan** **(vegetarian variant)**	All foods heated above 46 °C	Only vegetable foods not heated above 42 °C. Dried vegetable foods allowed. Fruits, vegetables, nuts, seeds, cereals, sprouted legumes.
**Fruitarian**	All foods of animal origin including milk and dairy products, eggs; legumes, cereals, vegetables, algae, mushrooms. Fruits and vegetables deriving from roots, flowers and leave and they are not the real fruit of the plant (strawberries, figs…)	Fresh or dry fruits (apple, pear, apricot, peach…), fruit vegetables (tomatoes, peppers, cucumbers…), high-fat fruits (olives, avocados).
**Windfall vegan (vegetarian variant)**	All foods that do not fall spontaneously from trees	Seeds or fruits that have fallen naturally from the trees/plants

^§^ propolis and royal jelly are derived from bees.

**Table 2 nutrients-14-03591-t002:** Auxological development and/or growth. Summary of findings for the main comparisons.

[Complementary Feeding Completely or Partially Free of Animal-Source Foods] Compared to [Balanced Omnivorous Diet] for [Different Auxological Development and/or Growth]					
**Patient or population:** [different auxological development and/or growth] **Setting:** Primary care **Intervention:** [complementary feeding completely or partially free of animal-source foods] **Comparison:** [balanced omnivorous diet]					
**Outcomes**	**№ of participants** **(studies)** **Follow-up**	**Certainty of the evidence** **(GRADE)**	**Relative effect** **(95% CI)**	**Anticipated absolute effects**	
				**Risk with [balanced omnivorous diet]**	*** Risk difference with [complementary feeding completely or partially free of animal-source foods]**
Risk of wasted grow thassessed with: % infant with major skin and muscle wasting follow-up: 2 years	106 (1 observational study) [42]	⨁⨁⨁◯ Moderate ^a^^,^^b^^,^^c^	RR 17.45 (2.39 to 127.38)	18 per 1000	289 more per 1000 (24 more to 2217 more)
Risk of stunted or wasted growth assessed with: % children with stunted growth	430 (1 observational study) [43]	⨁⨁⨁◯ Moderate ^c^^,^^d^	OR 13.97 (1.86 to 104.88)	6 per 1000	73 more per 1000 (5 more to 385 more)

* The risk in the intervention group (and its 95% confidence interval) is based on the assumed risk in the comparison group and the relative effect of the intervention (and its 95% CI). CI: confidence interval; OR: odds ratio; RR: risk ratio; ^a^. Ascertainment of the exposure; ^b^. A small sample population, inappropriately detailed diets, generically defined as “omnivorous” or “adequate” and “inadequate,”; ^c^. Wide 95% CI; ^d^. study design cross-sectional; single study. ⨁⨁⨁◯ means GRADE Working Group grades of evidence; High certainty: We are very confident that the true effect lies close to that of the estimate of the effect. Moderate certainty: We are moderately confident in the effect estimate; the true effect is likely to be close to the estimate of the effect, but there is a possibility that it is substantially different. Low certainty: Our confidence in the effect estimate is limited; the true effect may be substantially different from the estimate of the effect. Very low certainty: We have very little confidence in the effect estimate; the true effect is likely to be substantially different from the estimate of effect.

**Table 3 nutrients-14-03591-t003:** Psychomotor development. Summary of findings for the main comparisons.

[Complementary Feeding Completely or Partially Free of Animal-Source Foods] Compared to [Balanced Omnivorous Diet] for [Psychomotor Development That Is Significantly Different]					
**Patient or population:** [psychomotor development that is significantly different] **Setting:** Primary care **Intervention:** [complementary feeding completely or partially free of animal-source foods] **Comparison:** [balanced omnivorous diet]					
**Outcomes**	**№ of participants** **(studies)** **Follow-up**	**Certainty of the evidence** **(GRADE)**	**Relative effect** **(95% CI)**	**Anticipated absolute effects**	
				**Risk with [balanced omnivorous diet]**	*** Risk difference with [complementary feeding completely or partially free of animal-source foods]**
Psychomotor development assessed with: standardized psychomotor checklist (score)	106 (1 observational study) [42]	⨁⨁◯◯ Low ^a^^,^^b^	-	The mean psychomotor development was 0	−0.63 0 (0 to 0)
Psychomotor development assessed with: case report e case series	(7 observational studies) [43]	⨁⨁⨁◯ Moderate ^c^	10 cases of children aged 8–18 months: all of them exhibit severe neurological outcomes and growth deficits resulting from low vitamin B12 and vitamin D levels, with anemia, stunting, brain abnormalities, and demyelination. Cases with persistent outcomes are reported with no long-term follow-up data for any other disorders.		

* The risk in the intervention group (and its 95% confidence interval) is based on the assumed risk in the comparison group and the relative effect of the intervention (and its 95% CI). CI: confidence interval. ^a^. Ascertainment of the exposure; ^b^. A small sample population, inappropriately detailed diets, generically defined as “omnivorous” or “adequate” and “inadequate,”; ^c^. Wide 95% CI. ⨁⨁◯◯ and ⨁⨁⨁◯ means GRADE Working Group grades of evidence. High certainty: We are very confident that the true effect lies close to that of the estimate of the effect. Moderate certainty: We are moderately confident in the effect estimate; the true effect is likely to be close to the estimate of the effect, but there is a possibility that it is substantially different. Low certainty: Our confidence in the effect estimate is limited; the true effect may be substantially different from the estimate of the effect. Very low certainty: We have very little confidence in the effect estimate; the true effect is likely to be substantially different from the estimate of effect.

## Data Availability

Not applicable.

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
