# Peer review of "Do Vegetarian Diets Provide Adequate Nutrient Intake during Complementary Feeding? A Systematic Review"

_nutrients, 2022, doi:10.3390/nu14173591_

Round 1

Reviewer 1 Report

The authors of the manuscript entitled "Do vegetarian diets provide adequate nutrient intake during complementary feeding? A Systematic Review and Meta-Analysis" present a thorough systematic review of the effect of vegetarian and vegan diets during complementary feeding and developmental outcomes in infants. In general they find that the overall quality of evidence is low, but make a good case for the need for their safety to be established as well as the potential harms with respect to nutrient deficiencies. Some minor changes are required, though I would recommend a major shortening of the introduction.

Major comments:

·      My main comment is that the abstract and title do not align with the work presented. Firstly, it is clear that there were an insufficient number of relevant publications to perform a meta-analysis, so this should be removed from the title and details in the methods on meta-analytic strategies should be taken out. These are unnecessary.

·      The abstract does not make a significant enough statement about the risk of dietary deficiencies and instead focuses on prevention of non-communicable diseases, which is less relevant to the title and purpose of the study, and also harder to examine. The statements from the conclusion of the main manuscript are more in keeping with their findings, and some of these should make their way to the abstract. The fact is that animal-free complementary feeding strategies have not been shown to be safe, and the current best evidence suggests that the risk of critical micronutrient deficiencies or insufficiencies and growth retardation is high.

·      There have been some issues copying the manuscript into the template. Many sentences are missing their first letter of the first word, and Figure 1 and Table 1 are hard to read. There also seems to be some duplication within Table 2.

·      The introduction is far too long. It only need to be long enough to motivate the review, with the rest in the discussion. In particular, the discussion around the Mediterranean diet seems to be unnecessary. Though I agree about the point of having a high-quality omnivorous diet be the ideal comparison diet (rather than a poor-quality Western diet), no studies have actually done this and therefore this is a point better made in the discussion. For the same reasons I would take out the mention of the Mediterranean diet from the abstract.

Minor comments:

·      Mention in Table 1 that propolis/royal jelly are derived from bees.

·      “W” and “L” as growth outcomes are not defined – presumably this is weight and length?

Author Response

Dear Reviewer,

Thank you a lot for your comments and concerns about our paper. This helps a lot to improve it.

Here below please find our responses to your concerns.

Comments and Suggestions for Authors

The authors of the manuscript entitled "Do vegetarian diets provide adequate nutrient intake during complementary feeding? A Systematic Review and Meta-Analysis" present a thorough systematic review of the effect of vegetarian and vegan diets during complementary feeding and developmental outcomes in infants. In general they find that the overall quality of evidence is low, but make a good case for the need for their safety to be established as well as the potential harms with respect to nutrient deficiencies. Some minor changes are required, though I would recommend a major shortening of the introduction.

We thank the reviewer for this suggestion. We have shortened the introduction.

Major comments:

My main comment is that the abstract and title do not align with the work presented. Firstly, it is clear that there were an insufficient number of relevant publications to perform a meta-analysis, so this should be removed from the title and details in the methods on meta-analytic strategies should be taken out. These are unnecessary.

 Thanks for this interesting comment which allows us to detail the procedure of the meta-analysis. Although our meta-analyses were done on 1 study, they were informative because they highlighted some findings that were not clear from the study paper:

  1. the risk of insufficient growth
  2. no preventive effect on the risk of overweight/obesity

With regard to the opportunity and validity of the meta-analyses done on 1 study, although by definition meta-analysis applies to pooled data from 2 or more studies, it is common to find meta-analyses that include only 1 study, even in SRs Cochrane, for several reasons:

  1. it is formal respect for the protocol that provides for the meta-analysis of the results, even if the systematic review led to the selection of only one study;
  2. can give immediate evidence that there is only one study on the specific outcome and no more studies that cannot be merged;
  3. calculates effect measures not reported in the original study (is our case);
  4. offers a graphical representation of the results with the forest plot.

The abstract does not make a significant enough statement about the risk of dietary deficiencies and instead focuses on prevention of non-communicable diseases, which is less relevant to the title and purpose of the study, and also harder to examine. The statements from the conclusion of the main manuscript are more in keeping with their findings, and some of these should make their way to the abstract. The fact is that animal-free complementary feeding strategies have not been shown to be safe, and the current best evidence suggests that the risk of critical micronutrient deficiencies or insufficiencies and growth retardation is high.

We thank the Reviewer for his comments and valuable and appropriate suggestions. Based on his suggestions, we have made some changes to the article.

There have been some issues copying the manuscript into the template. Many sentences are missing their first letter of the first word, and Figure 1 and Table 1 are hard to read. There also seems to be some duplication within Table 2.

We thank you for the comments, and we have provided the figures in a better resolution size and have rechecked the tables. Table 2 is automatically created with the GRADEPRO program, which includes repeated headings. The data entered by us has no repetition. For any reading difficulties in the article due to format restrictions, all Tables and Figures can be viewed with better resolution in the Supplementary files.

The introduction is far too long. It only need to be long enough to motivate the review, with the rest in the discussion. In particular, the discussion around the Mediterranean diet seems to be unnecessary. Though I agree about the point of having a high-quality omnivorous diet be the ideal comparison diet (rather than a poor-quality Western diet), no studies have actually done this and therefore this is a point better made in the discussion. For the same reasons I would take out the mention of the Mediterranean diet from the abstract.

We thank the reviewer for the excellent suggestion and have implemented the mentioned corrections.

We have left in the abstract the reference to the Mediterranean Diet only as an easy to understand example, as MD is well known in the world.

Minor comments:

  • Mention in Table 1 that propolis/royal jelly are derived from bees.

According your request a note has been added to the table.

“W” and “L” as growth outcomes are not defined – presumably this is weight and length?

Thanks for noticing this little aspect. To us it means that the reviewer has really read our article carefully and we thank him/her for being so thorough. We defined Weight and Length

Submission Date

31 July 2022

Date of this review

08 Aug 2022 19:44:32

Date of this response

21 Aug 2022

Reviewer 2 Report

The manuscript entitled “Do vegetarian diets provide adequate nutrient intake during complementary feeding? A systematic review and meta analysis” by by simeone et al. provided interesting information. However, i have one question regarding this study. The author cited different studies in this paper but they didn’t provide the complete details for example in a study reference 53 by limeone et al. they presented the case report on rickets and author only mention about the symptoms in the present paper but didn't address the symptoms and reasons. 
So it is my suggestion to incorporate the relevant information in the manuscript for other studies as well (if needed).

Author Response

Dear Reviewer

Thanks a lot for your observations. We are honored you approved our work. We try to respond to your comments as below.

Comments and Suggestions for Authors

The manuscript entitled “Do vegetarian diets provide adequate nutrient intake during complementary feeding? A systematic review and meta analysis” by by Simeone et al. provided interesting information. However, i have one question regarding this study. The author cited different studies in this paper but they didn’t provide the complete details for example in a study reference 53 by Simeone et al. they presented the case report on rickets and author only mention about the symptoms in the present paper but didn't address the symptoms and reasons.

So it is my suggestion to incorporate the relevant information in the manuscript for other studies as well (if needed).

Thanks very much to Reviewer 2 for the suggestions. We have provided to better detail in the text, Section 3.3.2. the results of the studies lacking or unclear. The reference to the Table S3f for the description of each single study is also reported.

Submission Date

31 July 2022

Date of this review

15 Aug 2022 04:49:42

Date of this response

21 Aug 2022